# SNP-Based Heritability of Osteochondrosis Dissecans in Hanoverian Warmblood Horses

**DOI:** 10.3390/ani13091462

**Published:** 2023-04-25

**Authors:** Elisa Zimmermann, Ottmar Distl

**Affiliations:** Institute for Animal Breeding and Genetics, University of Veterinary Medicine Hannover (Foundation), 30559 Hannover, Germany

**Keywords:** equid, osteochondrosis, genetic parameters, genomic relationship matrices, SNP-based heritability, linkage disequilibrium

## Abstract

**Simple Summary:**

The heritability of a trait is the proportion of phenotypic variance explained via genetic variance. Prior to the advent of genomics, heritability was estimated using extensive pedigree analyses. With the availability of genome-wide genotyping arrays, an alternative method became available to estimate heritability using genomic relationship matrices derived from genotype data. We used approaches that consider patterns of linkage disequilibrium and relatedness to estimate heritability of osteochondrosis dissecans in Hanoverian Warmblood horses based on genotype data from SNP arrays and imputed genotype data. Taking into account linkage disequilibrium patterns and relatedness in the data, heritability estimates on the linear scale for fetlock-, hock- and stifle-OCD were 0.41–0.43, 0.62–0.63, and 0.23–0.25, respectively, with standard errors of 0.11–0.14. In summary, SNP-based approaches are able to capture a greater proportion of additive genetic variance than previous estimates based on pedigree data.

**Abstract:**

Before the genomics era, heritability estimates were performed using pedigree data. Data collection for pedigree analysis is time consuming and holds the risk of incorrect or incomplete data. With the availability of SNP-based arrays, heritability can now be estimated based on genotyping data. We used SNP array and 1.6 million imputed genotype data with different minor allele frequency restrictions to estimate heritabilities for osteochondrosis dissecans in the fetlock, hock and stifle joints of 446 Hanoverian warmblood horses. SNP-based heritabilities were estimated using a genomic restricted maximum likelihood (GREML) method and accounting for patterns of regional linkage disequilibrium in the equine genome. In addition, we employed GREML for family data to account for different degrees of relatedness in the study population. Our results indicate that we were able to capture a larger proportion of additive genetic variance compared to pedigree-based estimates in the same population of Hanoverian horses. Heritability estimates on the linear scale for fetlock-, hock- and stifle-osteochondrosis dissecans were 0.41–0.43, 0.62–0.63, and 0.23–0.25, respectively, with standard errors of 0.11–0.14. Accounting for linkage disequilibrium patterns had an upward effect on the imputed data and a downward impact on the SNP array genotype data. GREML for family data resulted in higher heritability estimates for fetlock-osteochondrosis dissecans and slightly higher estimates for hock-osteochondrosis dissecans, but had no effect on stifle-osteochondrosis dissecans. The largest and most consistent heritability estimates were obtained when we employed GREML for family data with genomic relationship matrices weighted through patterns of regional linkage disequilibrium. Estimation of SNP-based heritability should be recommended for traits that can only be phenotyped in smaller samples or are cost-effective.

## 1. Introduction

Osteochondrosis (OC) is one of the most important orthopaedic diseases of the juvenile horse [1]. Due to disturbances in enchondral ossification, damage to the subchondral bone is the reason for the formation of intraarticular osteochondral fragments and subchondral bone cysts. If osteochondral fragments occur, the disease is referred to as osteochondrosis dissecans (OCD). Osteochondrosis occurs at certain predilection sites. Joints frequently affected are the metacarpophalangeal/metatarsophalangeal (fetlock), tarsocrural (hock) and femoropatellar joints (stifle) [2]. Therefore, it is of utmost interest to evaluate genetic parameters for OCD as precisely as possible in order to take breeding measures.

The aetiology of OCD appears to be multifactorial with a relevant genetic contribution [3,4]. There have been estimations regarding the heritability of OCD based on pedigree [5,6,7,8,9,10,11,12,13,14,15,16,17,18,19,20,21] and genotyping data [22,23]. Those estimates are shown in a previous review [1] and supplemented with results of more recent studies shown in Appendix A.

Before the genomics era, estimates of heritability were based on pedigree data. The introduction of SNP arrays enabled the estimation of heritability based on genotyping data. Genome-based heritability estimates offer many advantages through eliminating the need to collect extensive pedigree data. Apart from the time-consuming data collection, analysis of pedigree data poses the risk of biased results due to incorrect, incomplete, or varying depth of pedigrees. Heritability estimates between populations may vary because of population history, gene frequency, or environmental exposures [23].

There are various approaches to estimating heritability based on genotyping data. The fraction of phenotypic variance that can be explained using variants that have been identified as causal variants through genome-wide association studies (GWAS) is named $h_{\mathrm{GWAS}}^{2}$. $h_{\mathrm{GWAS}}^{2}$ is limited because, for most complex diseases, only a small proportion of variants has already been identified [24]. $h_{\mathrm{SNP}}^{2}$ is the proportion of phenotypic variance explained using all SNPs on a genotyping platform. $h_{\mathrm{SNP}}^{2}$ is the upper limit for $h_{\mathrm{GWAS}}^{2}$ and can be a measurement of the proportion of already identified causal variants in the actual genetic variance of a trait [24]. The difference between $h_{\mathrm{GWAS}}^{2}$ and $h_{\mathrm{SNP}}^{2}$ is often referred to as missing heritability [25,26]. We want to estimate $h_{\mathrm{SNP}}^{2}$ and assess different methods using genomic REML algorithms (GREML). Heritability estimation methods based on genotyping data require certain assumptions regarding the population structure of the underlying population, indirect genetic effects, the presence of artificial or natural selection within the population, and linkage disequilibrium. These assumptions are specific for each population and trait and can severely bias the heritability estimates [27]. Different methods require certain assumptions [28]. The aim of heritability estimation using SNP array or Beadchip data is to approach h^2^, which is the actual narrow sense heritability of a trait [29].

We estimated $h_{\mathrm{SNP}}^{2}$ using GCTA GREML [30], which is a single-component model to estimate heritability based on a genomic relationship matrix (GRM) and unrelated individuals [31]. As this approach is very sensitive to patterns of linkage disequilibrium (LD) [32], we used a similar single-component approach that is implemented in the software LDAK and weights SNP effect sizes according to regional LD patterns to construct the GRM [32]. LD describes the non-random association of alleles at two or more loci. LD varies because of factors such as population history, natural or artificial selection, mutation, and other forces that cause changes in allele frequency [33]. It can cause upwards biased estimates of heritability due to repeated tagging of SNPs [34].

As Beadchip arrays are usually based on common SNPs, we want to use imputed SNPs for heritability estimation to capture the effects of more causal variants [27,35,36,37]. However, it is recommendable to prune for minor allele frequency (MAF) because rare variants are imputed less accurately [36,38].

In a previous study on osteochondrosis in horses, SNP-based heritability for osteochondrosis in the hock was estimated in a population of 479 North American Standardbred. Horses using REML analysis in GCTA and LDAK with the weighted GRM in an imputed data set with ~1.25 million SNPs. The OCD frequency in this study population was 0.27. The analyses were repeated using a smaller study population, with individuals pruned for relatedness at 0.25. SNP-based estimates seemed to be biased upwards via LD, which implies the need to account for LD in heritability estimations in horse populations [20].

Zaitlen et al. [24] proposed a method to estimate heritability based on a population with different degrees of kinship that avoids the need to remove closely related individuals from the study population. This method is implemented in GCTA and is known as GREML analysis for family data. It provides estimations of SNP-based heritability in family data as well as narrow sense heritability $h_{f}^{2}$, which enables the quantification of genetic effects resulting from kinship and, thus, enables the detection of higher amounts of heritability [24]. This method has not yet been used in horse populations with very specific relatedness structures.

The aim of this work is to estimate the heritability of the trait OCD in the fetlock, hock, and stifle joints based on SNP data. We conducted a GREML analysis, an LDAK analysis using an LD-weighted GRM based on unrelated individuals, and a GREML analysis for family data using two simultaneously constructed GRMs for individuals with different relatedness structures. Beside the effects of the different GRMs, we will observe the effects of imputation and the different MAF restrictions, comparing the heritability estimates with previous pedigree-based analyses using a similar study sample [5].

## 2. Materials and Methods

### 2.1. Animals

The horses included in this study were a subset of the study population previously analyzed by Hilla et al. [5]. For the present study, 446 four-year-old Hanoverian warmblood horses were included. The inclusion criteria were as follows: only one horse per sire and maternal sire was allowed, either as a control or a case. The controls and cases were randomly distributed among the sires and maternal sires. The control horses had to be free of all diseases found during the veterinary health examination for pre-selection at auctions, at licensing, or during the purchase examination. The cases were horses with OCD only and free of any other disease recorded in the veterinary health check. The veterinary health check included clinical and radiographic examination of all four limbs. Only osteochondral fragments at the specific predilection sites of the fetlock, hock, and stifle joints were classified as OCD [5,39]. Osteochondral fragments plantar in the fetlock joints and at the insertion sites of the short sesamoid ligaments at the proximal phalanx of the hindlimbs were classified as plantar and dorsodistal fragments of the fetlock joints; thus, there were not considered OCD. Distal and proximal interphalangeal joints, fetlock joints, hock joints, and stifle joints were evaluated for contour changes stemmed from periarticular osteophytes or exostoses and for a narrow or absent joint space. These changes were classified as osteoarthroses. Radiographic changes in the shape, symmetry, contour, and structure of the navicular bone and the shape, size, number, and location of the canales sesamoidales were scored on a scale of 1–4 [40]. Only horses with a score of 1 were considered free of radiographic changes to the navicular bone. Horses with the presence of a sidebone were also scored as not being free of radiographic changes. After removing all horses affected by diseases other than OCD, the data set was filtered for cases and controls. The strict inclusion criteria resulted in the final data set comprising 446 horses. Traits were encoded as 0/1 variates for each joint. We did not consider an overall score for OCD because genetic correlations of OCD between the different joints were moderately negative (fetlock-OCD with hock- and stifle-OCD: −0.12 and −0.18) or moderately positive (hock-OCD with stifle-OCD: 0.17) [5].

The phenotypic and genotype data were provided by the Association of Hanoverian Warmblood breeders (Hannoveraner Verband e.V., Verden, Germany). The frequencies of OCD were 0.2489 (*n* = 111), 0.3139 (*n* = 140), and 0.0291 (*n* = 13) in the fetlock, hock, and stifle joints, respectively. Relationships expressed through the contingency coefficient between the frequencies of fetlock, hock, and stifle OCD were close to zero because 96, 125, and 11 horses represented the sole cases of fetlock, hock, and stifle joint OCD, respectively.

### 2.2. Methods

Genome-wide genotyping data was obtained using the GGP Equine (71 589 SNPs) genotyping array. Descriptive statistics of the population were calculated with SAS, version 9.4 (Statistical Analysis System, Cary, NC, USA, 2023). The SNP data have been imputed to 1,617,270 SNPs with an information score of 0.95 using BEAGLE 5.4 [41] and publicly available whole genome sequencing data for horses (Appendix A). Subsequently, the imputed and non-imputed data sets were pruned at MAFs of 0.01, 0.025, or 0.05 using PLINK 1.9 [42,43], resulting in six different data sets. Using all six data sets, heritabilities for OCD of the fetlock, hock, and stifle joints were estimated using the GREML analysis implemented in GCTA (genome-wide complex trait analysis) 1.94.1 [30] with one GRM [44], resulting in SNP-based heritability ($h_{\mathrm{SNP}}^{2}$). Subsequent estimations were performed using the LD-weighted genomic relatedness matrix as implemented in LDAK 5.2 [45] and the integrated REML analysis [32], resulting in estimations of $h_{\mathrm{SNPw}}^{2}$. Using the GREML analysis for family data with two GRMs simultaneously, based on all pairs of individuals and related individuals [24] implemented in GCTA 1.94.1 [30], we estimated $h_{f}^{2}$. The GRM based on all pairs of individuals captured information on the sharing of causal variants tagged using SNPs. The second GRM considered only individuals who were identical-by-state above a certain threshold (0.05) and, consequently, only related individuals. Hence, it captured information on shared causal variants that could not be tagged using SNPs [24,29]. Both GRMs fitted into a mixed linear model and were supposed to provide estimates of $h_{\mathrm{SNP}-\mathrm{all}-\mathrm{pairs}}^{2}$ from the first GRM and the missing heritability $h_{\mathrm{SNP}-\mathrm{related}}^{2}$ from the second GRM. Those values were summed up to $h_{f}^{2}$ [24].

We obtained heritability estimates ($h_{\mathrm{fw}}^{2}$ and $h_{\mathrm{SNP}-\mathrm{all}-\mathrm{pairs}-w}^{2}$) by implementing the LD-weighted genomic relationship matrix estimated with LDAK [32] in the GREML analysis for family data [24]. Sex was included in all analyses as a covariate. As we used 0/1-data, all heritability estimates were transformed onto the liability scale using the prevalence option of GCTA. The study design for heritability estimations is illustrated in Appendix A.

## 3. Results

The results of our heritability estimates for osteochondrosis in the fetlock joint are given in Table 1. Additionally, estimates for $h_{\mathrm{SNP}-\mathrm{all}-\mathrm{pairs}}^{2}$ and $h_{\mathrm{SNP}-\mathrm{all}-\mathrm{pairs}-w}^{2}$ are given in Appendix A. The SNP-based heritabilities estimated with GREML revealed that the heritability estimates decreased in the imputed data set compared with the original data set. The SNP-based heritabilities estimated with GREML and LDAK for fetlock-OCD differed in several aspects. Accounting for regional LD patterns increased heritability estimates for the imputed genotype data but slightly decreased heritability estimates for the original data sets. Heritability estimates using GREML analysis for family data resulted in higher estimates for both data sets, the original and imputed genotype data, as well as when regional LD patterns were considered. The effects of using different MAFs had only small effects when we used LD-weighted genomic relatedness matrices with LDAK. Standard errors for heritability estimates using GREML analysis for family data slightly increased from 0.11–0.12 to 0.13–0.14 on the linear scale.

After transformation onto the liability scale, we obtained fairly high estimates for heritability and their standard errors.

When comparing the GREML analysis for the original and imputed data sets, the same trends were observed for the heritability estimates for hock- and fetlock-OCD (Table 2, Appendix A). However, the increase in heritability estimates was much smaller when GREML analysis was applied to family data than to fetlock-OCD. The consistency and magnitude of the heritability estimates were highest when we used GREML analysis for family data with LDAK.

The most consistent heritability estimates were obtained for stifle-OCD for the analysis accounting for family data and LD patterns for both data sets (the original and imputed genotype data) (Table 3, Appendix A). The effect of family structure on heritability estimates was small, while LD patterns had slightly larger effects. In general, differences between the different approaches were relatively small. Transformation onto the liability scale gave meaningless estimates >1 due to the low frequency of cases.

## 4. Discussion

According to the findings of previous studies, it seems to be recommendable to account for linkage disequilibrium when estimating heritability based on SNP data [20,29,31,32,34,46]. Horses have long-range linkage disequilibria, which is why SNPs can show effects of a risk variant as far away as 1 Mb [47]. Additionally, LD is higher within breeds than across breeds [48], which is important since population data are usually used for heritability estimations. In general, REML-based estimates, such as those obtained from GREML analysis in GCTA, are sensitive to patterns of LD [32]. The linkage disequilibrium between SNPs is used to create the GRM and the LD between SNPs; causal variants can cause bias in heritability estimation [32,36]. As the intensity of linkage disequilibrium varies regionally along the genome, LDAK weights the SNPs according to local patterns of LD [32]. While we cannot observe a large impact of LD using our original data set, we see slight differences between GREML analysis and LDAK analysis in the imputed data set. The difference between heritability estimates increases with increasing MAF restrictions, which is attributable to the fact that less genetic variation is captured with SNPs when lower frequencies are recorded. Additionally, allele frequency and linkage disequilibrium are dependent on each other [49], which explains why the estimations conducted with LDAK are able to compensate changes in MAF. We assume that linkage disequilibrium does not play a major role in our study population. One possible explanation could be that we included many individuals with diverse LD structures, meaning that they outweighed each other in our analysis.

One cause of undetected heritability could be that rare variants, and eventually even variants with large effects, may not be mapped on the available genotyping arrays that mainly include common SNPs [36]. Therefore, it is recommended to perform heritability estimations on imputed data sets [29]. To capture as much variation as possible, we imputed our Beadchip data to 1,617,270 SNPs, which corresponds to the recommendations given by Evans et al. [29] for heritability estimations. When comparing the results of the original and imputed data sets, we observe for all traits analyzed the most consistent estimates when family data and LD patterns are accounted for. Even the differences between the original and imputed data shrink or are no longer present. In the present data set, imputation had no or very little effect on SNP-based heritabilities; thus, we were unable to detect variation due to rare alleles.

The single-component analyses in GCTA and LDAK calculated GRM based on the available SNP data to subsequently estimate heritability. For those analyses, it was recommended to prune for relatedness to eradicate bias caused by common environmental or other non-additive genetic effects [29]. The resulting unrelated individuals are by definition distantly related individuals because they share distant ancestors [50]; however, they are assumed to provide random genetic variance [28]. The need for pruning for relatedness arises from the model assumption in the GREML analysis that all measured genetic effects are direct effects. If related individuals were included, the indirect genetic effects between those individuals would be counted as direct effects and, thus, inflate heritability estimates [28]. Indirect genetic effects may result from genetic maternal effects [28]. The idea of the GREML analysis for family data was to find a way to circumvent pruning for relatedness in a study population and, thus, ensure a larger study population, which in turn should lead to lower standard errors. Additionally, the GREML analysis for family data estimates $h_{f}^{2}$ and, thus, is able to capture higher heritability [24]. While $h_{f}^{2}$ provides an unbiased estimate of the heritability of the trait, the proportions of the single components do not always seem to be assessed correctly [24,29]. We only observed this phenomenon in the imputed data set for stifle-OCD when we employed GREML for family data without an LD-weighted genomic relationship matrix. In all other analyses, we could not observe imbalanced contributions to the heritability estimates resulting from the two different GRMs. The most likely reason for this issue is the very low frequency of cases for stifle-OCD.

In our analyses for fetlock-OCD, we can confirm that we detected higher estimates of heritability with GREML for family data than with the single-component REML algorithms, whereas for hock- and stifle-OCD the increase in heritability estimates was rather small. We assume that in our population a significant amount of heritability for fetlock-OCD may be due to indirect genetic effects that are captured examining the genetic effects between individuals with varying degrees of relatedness. This is the first analysis with GREML for family data that has been performed in a horse population. The most consistent heritability estimates were obtained using GREML for family data and an LD-weighted genomic relationship matrix for the original and imputed genotype data. With frequencies of cases closer to 0.5 in the population under study, differences between the original and imputed data sets diminished. However, we have to note that GREML for family data is designed for human populations with their specific relatedness structure and significantly larger available data sets [24,31]. While in a human population, full siblings are common, in horse population, full-siblings are uncommon.

Since OCD is defined as a binary trait, all results of the REML analyses have been transformed onto the liability scale as recommended [30,51,52,53]. In agreement with previous studies, upward bias may occur, particularly when estimates on the linear scale are high and more frequencies deviate from 0.5 [11,12].

The present study used data from Hilla et al. [5]. We selected horses as representatively as possible and avoided including closely related animals, such as paternal half-siblings and maternal sire half-siblings. The results obtained from the present study allow us to assume that analyses using GREML for family data and an LD-weighted genomic relationship matrix result in higher heritability estimates compared to estimates based on pedigree data ($h_{\mathrm{ped}}^{2}$) in a Hanoverian Warmblood horse population. However, larger genotype data sets should be available to reach lower standard errors. Nevertheless, heritability estimations based on SNP-based methods may give reliable results even in much smaller data sets compared to pedigree-based estimates.

Similar results were reported for hock-OCD in a population of North American Standardbred horses [20]. Compared to our results, this previous study showed larger increases in heritability estimates when taking into account LD patterns compared to the standard GRM. Thus, we assume stratification based on families and breed history may have contributed to this result. In addition, the LDAK version used by McCoy et al. [20,32] was a less improved version of the software, which could have had an effect on the results. Heritability estimates are specific for populations because of different familial structures and selective signatures in the genome [1,33], which may also contribute to the difference in our results. In the present study, standard errors on the linear scale were at 0.11–0.14, resulting in 95% confidence intervals from ±0.22 to ±0.27, while standard errors on the linear scale were at 0.12 and 0.16 in the previous study on US Standardbreds.

In summary, we recommend the use of GREML for family data with an LD-weighted genomic relatedness matrix to estimate heritabilities, particularly for traits which are difficult or very costly to record. Due to restrictions in sampling and varyingly strong LD patterns in populations, the approach as provided by LDAK should be implemented in the estimation procedure. The pursuit of more precise heritability estimates is worthwhile means of achieving estimated breeding values with higher reliabilities and a higher selection response in health traits.

## 5. Conclusions

Estimation of heritabilities based on SNP arrays is recommended because reasonably high accuracy of estimates can be achieved in smaller samples compared to pedigree-based studies with similar sample sizes. The use of genomic REML analysis for family data with LD-weighted genomic relationship matrices allows the capture of most of the additive genetic variance and provides the most consistent estimates at different MAFs. The present study yielded higher heritability estimates with reasonable standard errors than a previous study for the same population. Further studies with larger data sets should be performed to validate these results.

## Figures and Tables

**Table 1 animals-13-01462-t001:** Heritability estimates with their standard errors (h^2^ ± SE) for OCD in fetlock joint estimated with GCTA GREML, LDAK, and GCTA GREML for family data and GCTA GREML for family data and a LD-weighted genomic relationship matrix; the original and imputed SNP data and, at minor allele frequencies (MAF) of 0.01, 0.025 and 0.05, is included with transformation to the liability scale (Obs = observed scale, Liab = liability scale).

Approach	Data Set	MAF 0.01		MAF 0.025		MAF 0.05	
		Obs	Liab	Obs	Liab	Obs	Liab
GREML ($h_{\mathrm{SNP}}^{2}\pm \mathrm{SE}$)	Original	0.34 ± 0.12	0.64 ± 0.22	0.33 ± 0.12	0.61 ± 0.22	0.32 ± 0.12	0.60 ± 0.22
	Imputed	0.31 ± 0.11	0.58 ± 0.21	0.30 ± 0.11	0.56 ± 0.20	0.28 ± 0.11	0.53 ± 0.20
LDAK ($h_{\mathrm{SNPw}}^{2}\pm \mathrm{SE}$)	Original	0.33 ± 0.12	0.61 ± 0.22	0.33 ± 0.12	0.61 ± 0.22	0.32 ± 0.12	0.60 ± 0.22
	Imputed	0.34 ± 0.12	0.64 ± 0.23	0.34 ± 0.12	0.63 ± 0.22	0.33 ± 0.12	0.62 ± 0.22
GREML fam ($h_{f}^{2}\pm \mathrm{SE}$)	Original	0.43 ± 0.14	0.80 ± 0.26	0.42 ± 0.14	0.78 ± 0.26	0.44 ± 0.14	0.81 ± 0.26
	Imputed	0.41 ± 0.13	0.76 ± 0.24	0.40 ± 0.13	0.74 ± 0.24	0.38 ± 0.13	0.71 ± 0.23
GREML fam LD-weighted ($h_{\mathrm{fw}}^{2}\pm \mathrm{SE})$	Original	0.41 ± 0.14	0.76 ± 0.26	0.41 ± 0.14	0.77 ± 0.26	0.41 ± 0.14	0.76 ± 0.26
	Imputed	0.43 ± 0.14	0.79 ± 0.26	0.42 ± 0.14	0.79 ± 0.26	0.43 ± 0.14	0.81 ± 0.26

**Table 2 animals-13-01462-t002:** Heritability estimates with their standard errors (h^2^ ± SE) for osteochondrosis dissecans in hock joint estimated with GCTA GREML, LDAK, and GCTA GREML for family data and GCTA GREML for family data; a LD-weighted genomic relationship matrix, with the original and imputed SNP data and at minor allele frequencies (MAF) of 0.01, 0.025 and 0.05, is also included with transformation to the liability scale (Obs = observed scale, Liab = liability scale).

Approach	Data Set	MAF 0.01		MAF 0.025		MAF 0.05	
		Obs	Liab	Obs	Liab	Obs	Liab
GREML ( $h_{\mathrm{SNP}}^{2}\pm \mathrm{SE}$)	Original	0.60 ± 0.11	1.02 ± 0.19	0.60 ± 0.11	1.01 ± 0.18	0.57 ± 0.11	0.98 ± 0.18
	Imputed	0.54 ± 0.10	0.93 ± 0.18	0.52 ± 0.10	0.90 ± 0.18	0.50 ± 0.10	0.85 ± 0.17
LDAK ($h_{\mathrm{SNPw}}^{2}\pm \mathrm{SE}$)	Original	0.59 ± 0.11	1.01 ± 0.19	0.59 ± 0.11	1.00 ± 0.19	0.58 ± 0.11	1.00 ± 0.19
	Imputed	0.62 ± 0.11	1.06 ± 0.19	0.61 ± 0.11	1.05 ± 0.19	0.60 ± 0.11	1.03 ± 0.19
GREML fam ($h_{f}^{2}\pm \mathrm{SE}$)	Original	0.62 ± 0.12	1.07 ± 0.21	0.63 ± 0.12	1.09 ± 0.21	0.63 ± 0.12	1.09 ± 0.21
	Imputed	0.57 ± 0.12	0.97 ± 0.21	0.56 ± 0.12	0.95 ± 0.21	0.53 ± 0.12	0.91 ± 0.20
GREML fam LD-weighted ( $h_{\mathrm{fw}}^{2}\pm \mathrm{SE}$)	Original	0.63 ± 0.12	1.08 ± 0.21	0.63 ± 0.12	1.08 ± 0.21	0.62 ± 0.12	1.07 ± 0.21
	Imputed	0.63 ± 0.12	1.09 ± 0.21	0.62 ± 0.12	1.07 ± 0.21	0.63 ± 0.12	1.07 ± 0.21

**Table 3 animals-13-01462-t003:** Heritability estimates with their standard errors (h^2^ ± SE) for osteochondrosis dissecans in stifle joint estimated with GCTA GREML, LDAK, AND GCTA GREML for family data and GCTA GREML for family data; a LD-weighted genomic relationship matrix, with the original and imputed SNP data and at minor allele frequencies (MAF) of 0.01, 0.025 and 0.05, is also included with transformation to the liability scale (Obs = observed scale, Liab = liability scale).

Approach	Data Set	MAF 0.01		MAF 0.025		MAF 0.05	
		Obs	Liab	Obs	Liab	Obs	Liab
GREML ( $h_{\mathrm{SNP}}^{2}\pm \mathrm{SE}$)	Original	0.25 ± 0.11	1.60 ± 0.69	0.24 ± 0.11	1.55 ± 0.68	0.23 ± 0.11	1.47 ± 0.68
	Imputed	0.19 ± 0.10	1.25 ± 0.64	0.17 ± 0.10	1.11 ± 0.62	0.16 ± 0.10	1.04 ± 0.61
LDAK ($h_{\mathrm{SNPw}}^{2}\pm \mathrm{SE}$)	Original	0.23 ± 0.11	1.50 ± 0.69	0.23 ± 0.11	1.49 ± 0.68	0.23 ± 0.11	1.47 ± 0.68
	Imputed	0.21 ± 0.11	1.37 ± 0.68	0.20 ± 0.11	1.29 ± 0.67	0.20 ± 0.10	1.27 ± 0.67
GREML fam ($h_{f}^{2}\pm \mathrm{SE}$)	Original	0.26 ± 0.12	1.66 ± 0.75	0.24 ± 0.12	1.55 ± 0.75	0.23 ± 0.12	1.49 ± 0.74
	Imputed	0.24 ± 0.11	1.53 ± 0.70	0.22 ± 0.11	1.40 ± 0.70	0.23 ± 0.11	1.45 ± 0.70
GREML fam LD-weighted ( $h_{\mathrm{fw}}^{2}\pm \mathrm{SE}$)	Original	0.25 ± 0.12	1.63 ± 0.75	0.25 ± 0.12	1.63 ± 0.75	0.25 ± 0.12	1.60 ± 0.75
	Imputed	0.23 ± 0.12	1.48 ± 0.74	0.23 ± 0.11	1.48 ± 0.73	0.23 ± 0.11	1.45 ± 0.73

## Data Availability

Restrictions apply to the availability of these data. Data and genotypes were provided by the Association of Hanoverian Warmblood breeders (Hannoveraner Verband e.V., Verden, Germany) and are available on reasonable request from the authors with the permission of the Association of Hanoverian Warmblood breeders.

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
