# Peer review of "SNP-Based Heritability of Osteochondrosis Dissecans in Hanoverian Warmblood Horses"

_animals, 2023, doi:10.3390/ani13091462_

Round 1

Reviewer 1 Report

2.1 Animals could be improved. In L 270, authors state that horses were selected as representative as possibile, and avoiding close relatives. I think these criteria must be better described in M&M.

The OCD frequency in the fetlock and hock joints were approximately like in a 3:1 control:case study, whilst the OCD frequency in the stifle joint was very low (13/446): I know that the prevalence of this disease is lower, but I also think that the available dataset was large, so the subsetting seem strange. As a result, when the heritability estimates have been transformed to the continuous liability scale they exceded 1.

Author Response

Reviewer 1:

We thank the reviewer for the valuable comments and recommendations to improve our manuscript. We revised our manuscript accordingly.

Open Review

Quality of English Language

( ) English very difficult to understand/incomprehensible
( ) Extensive editing of English language and style required
( ) Moderate English changes required
( ) English language and style are fine/minor spell check required
(x) I am not qualified to assess the quality of English in this paper

Yes

Can be improved

Must be improved

Not applicable

Does the introduction provide sufficient background and include all relevant references?

(x)

( )

( )

( )

Are all the cited references relevant to the research?

(x)

( )

( )

( )

Is the research design appropriate?

(x)

( )

( )

( )

Are the methods adequately described?

( )

(x)

( )

( )

Are the results clearly presented?

(x)

( )

( )

( )

Are the conclusions supported by the results?

(x)

( )

( )

( )

2.1 Animals could be improved. In L 270, authors state that horses were selected as representative as possibile, and avoiding close relatives. I think these criteria must be better described in M&M.

Amended:

Lines 125-145

The inclusion criteria were as follows: only one horse per sire and maternal sire was allowed, either as a control or a case. The controls and cases were randomly distributed among the sires and maternal sires. The control horses had to be free of all diseases found during the veterinary health examination for pre-selection at auctions, at licensing, or during the purchase examination. The cases were horses with OCD only and free of any other disease recorded in the veterinary health check. The veterinary health check included clinical and radiographic examination of all four limbs. Only osteochondral fragments at the specific predilection sites of the fetlock, hock, and stifle joints were classified as OCD [5]. Osteochondral fragments plantar in the fetlock joints and at the insertion sites of the short sesamoid ligaments at the proximal phalanx of the hindlimbs were classified as plantar and dorsodistal fragments of the fetlock joints and were not considered OCD. Distal and proximal interphalangeal joints, fetlock joints, hock joints, and stifle joints were evaluated for contour changes caused by periarticular osteophytes or exostoses and for a narrow or absent joint space. These changes were classified as osteoarthroses. Radiographic changes in the shape, symmetry, contour, and structure of the navicular bone and the shape, size, number, and location of the canales sesamoidales were scored on a scale of 1-4. Only horses with a score of 1 were considered free of radiographic changes of the navicular bone. The presence of a lateral bone was also scored as not free of radiographic changes. After removing all horses affected by diseases other than OCD, the data set was filtered for cases and controls. The strict inclusion criteria resulted in the final data set comprising 446 horses.

The OCD frequency in the fetlock and hock joints were approximately like in a 3:1 control:case study, whilst the OCD frequency in the stifle joint was very low (13/446): I know that the prevalence of this disease is lower, but I also think that the available dataset was large, so the subsetting seem strange. As a result, when the heritability estimates have been transformed to the continuous liability scale they exceded 1.

Amended:

Lines 146-150:

The frequencies of osteochondrosis dissecans (OCD) were 0.2489 (n=111), 0.3139 (n=140), and 0.0291 (n=13) in the fetlock, hock and stifle joint, respectively. Relationships expressed by the contingency coefficient between the frequencies of fetlock, hock and stifle OCD were close to zero, because 96, 125 and 11 horses were sole cases for fetlock, hock and stifle joint OCD.

Reviewer 2 Report

I found this an interesting study to read and it was generally well written and explained. It might interest the horse community and it can promote this technique/methodology further.

My main remark is on the selection of the horses for this study and on the definition of the phenotype. This is not at all described.

L1: Title: I guess it is not only OCD but also OC? That should be made more clear in the title but also in the paper. I guess OCD and OC are not synonyms?

L8  Simple summary:  I would suggest to give some numbers (results or estimates of heritability) in this paragraph

L18   I do not really understand what you mean with “the risk of confounding with incorrect data” when applying pedigree based analysis.

L79   How did you achieve these “unrelated” individuals. Can you really compare this with human data? What is the criterion for unrelatedness?

L91   “estimated” can be dropped

L116   I find the description of the chosen set of animals much too vague. Cfr L79 
If I’m right over 7000 horses were in the 2013 study but now you end up with 446. How did you select these horses?

L121   Nothing is said about the way OC was categorized. Is this a 0/1 trait for each joint? Include the relevant ref here.  Also, is OC=OCD or did you treat as being the same?
Also, when treating a horse as an OC control for one joint, it could possibly be a case for another joint? Could you explain how you handle this king of horses?  Would it be feasible to make an overall-OC-score too?

L127 and L128    I’m unsure how I do have to understand this. Which whole genome sequences where used? All breeds or only Warmbloods?

L134    When we do not know how horses were selected/rejected (maybe also on relatedness) it is difficult to judge the value of including the GRM in the family-based approach?

L201   The sentence “In general, …..”, is weird.    GCTA is not a REML-based estimate I would say. It’s a software package

L222   So could you conclude that imputing is not that valuable?

L225   und => and

L228   per definition    by definition?

L232   I do not really understand what you mean with indirect genetic effects?

L265    First time that you talk about the phenotype as a binary trait. Should be done in Materials and methods

L269   Also, you discuss here how you selected animals but this needs to be described in the M&M

L283 …which may also could have had….sounds very heavy….which could have had….

L241 to L259    You discuss contributions to the estimates but I do not see where we can find these results. h2SNPf  is something you introduce her but was not mentioned before. This makes this part of the discussion a bit confusing.

L296    Conclusions

I’m not convinced that a pedigree based approach would yield lower accuracy. Maybe you can test it on your data and run a pedigree-based analysis.

I think however that the fact that imputation is maybe not needed and that family based data can be used is more important for the horse community.

Supplementary figure 1.

The complete figure is not really needed. The number of retained SNP’s in original and imputed datasets, after pruning for MAF, is certainly informative but the horizontal, colored bars are not needed.

Table 1:  Estimates of heritability for equine osteochondrosis reported in previous studies

The table does not seem to be very complete. This are not all previous studies. But I can understand that you do not want to write a review paper (again) but than you need to give an explanation why these studies and not other studies.

Author Response

Reviewer 2:

We thank the reviewer for the valuable comments and recommendations to improve our manuscript. We revised our manuscript accordingly.

Open Review

Quality of English Language

( ) English very difficult to understand/incomprehensible
( ) Extensive editing of English language and style required
( ) Moderate English changes required
(x) English language and style are fine/minor spell check required
( ) I am not qualified to assess the quality of English in this paper

Yes

Can be improved

Must be improved

Not applicable

Does the introduction provide sufficient background and include all relevant references?

(x)

( )

( )

( )

Are all the cited references relevant to the research?

( )

(x)

( )

( )

Is the research design appropriate?

(x)

( )

( )

( )

Are the methods adequately described?

( )

( )

(x)

( )

Are the results clearly presented?

( )

(x)

( )

( )

Are the conclusions supported by the results?

( )

( )

(x)

( )

Comments and Suggestions for Authors

I found this an interesting study to read and it was generally well written and explained. It might interest the horse community and it can promote this technique/methodology further.

My main remark is on the selection of the horses for this study and on the definition of the phenotype. This is not at all described.

L1: Title: I guess it is not only OCD but also OC? That should be made more clear in the title but also in the paper. I guess OCD and OC are not synonyms?

Amended:

Lines 125-145

The inclusion criteria were as follows: only one horse per sire and maternal sire was allowed, either as a control or a case. The controls and cases were randomly distributed among the sires and maternal sires. The control horses had to be free of all diseases found during the veterinary health examination for pre-selection at auctions, at licensing, or during the purchase examination. The cases were horses with OCD only and free of any other disease recorded in the veterinary health check. The veterinary health check included clinical and radiographic examination of all four limbs. Only osteochondral fragments at the specific predilection sites of the fetlock, hock, and stifle joints were classified as OCD [5]. Osteochondral fragments plantar in the fetlock joints and at the insertion sites of the short sesamoid ligaments at the proximal phalanx of the hindlimbs were classified as plantar and dorsodistal fragments of the fetlock joints and were not considered OCD. Distal and proximal interphalangeal joints, fetlock joints, hock joints, and stifle joints were evaluated for contour changes caused by periarticular osteophytes or exostoses and for a narrow or absent joint space. These changes were classified as osteoarthroses. Radiographic changes in the shape, symmetry, contour, and structure of the navicular bone and the shape, size, number, and location of the canales sesamoidales were scored on a scale of 1-4. Only horses with a score of 1 were considered free of radiographic changes of the navicular bone. The presence of a sidebone was also scored as not free of radiographic changes. After removing all horses affected by diseases other than OCD, the data set was filtered for cases and controls. The strict inclusion criteria resulted in the final data set comprising 446 horses.

L8  Simple summary:  I would suggest to give some numbers (results or estimates of heritability) in this paragraph

Amended:

Line 14-16

Taking into account linkage disequilibrium patterns and relatedness in the data, heritability estimates on the linear scale for fetlock-, hock- and stifle-OCD were 0.41-0.43, 0.62-0.63, and 0.23-0.25, respectively, with standard errors of 0.11-0.14.

L18   I do not really understand what you mean with “the risk of confounding with incorrect data” when applying pedigree based analysis.

Comment: data are usually not equally informative for each animal and pedigree errors also occur.

Amended:

Line 20-21:

and holds the risk of incorrect or incomplete data.

L79   How did you achieve these “unrelated” individuals. Can you really compare this with human data? What is the criterion for unrelatedness?

Amended:

Line 125-126

The inclusion criteria were as follows: only one horse per sire and maternal sire was allowed, either as a control or a case.

L91   “estimated” can be dropped

Amended:

Line 97:

In a previous study on osteochondrosis in the horse, SNP-based heritability for osteochondrosis

L116   I find the description of the chosen set of animals much too vague. Cfr L79 
If I’m right over 7000 horses were in the 2013 study but now you end up with 446. How did you select these horses?

Amended:

See above. Lines 125-145

L121   Nothing is said about the way OC was categorized. Is this a 0/1 trait for each joint? Include the relevant ref here.  Also, is OC=OCD or did you treat as being the same?
Also, when treating a horse as an OC control for one joint, it could possibly be a case for another joint? Could you explain how you handle this king of horses?  Would it be feasible to make an overall-OC-score too?

Comment: OC not equal to OCD. Trait definition followed Hilla & Distl-Papers. We did not consider an overall score. Looking on genetic correlations between these traits, this does not seem useful.

Amended:

See above and

Lines 145-147:

Traits were encoded as 0/1-variates for each joint.

Line 146-149:

We did not consider an overall score for OCD, because genetic correlations of OCD between the different joints were moderately negative (fetlock-OCD with hock- and stifle-OCD: -0.12 and -0.18) or moderately positive (hock-OCD with stifle-OCD: 0.17) [5].

L127 and L128    I’m unsure how I do have to understand this. Which whole genome sequences where used? All breeds or only Warmbloods?

Amended:

Line 162:

publicly available whole genome sequencing data of horses (Supplementary Table S2).

L134    When we do not know how horses were selected/rejected (maybe also on relatedness) it is difficult to judge the value of including the GRM in the family-based approach?

Comment: see above.

L201   The sentence “In general, …..”, is weird.    GCTA is not a REML-based estimate I would say. It’s a software package

Amended:

Line 246-247:

In general, REML-based estimates such as those obtained from GREML analysis in GCTA are sensitive to patterns of LD

L222   So could you conclude that imputing is not that valuable?

Amended:

Line 269-271:

In the present data set, imputation had no effect or very small effects on SNP-based heritabilities, so we were unable to detect variation due to rare alleles.

L225   und => and

Line 272:

The single component analyses in GCTA and LDAK calculate

Amended:

L228   per definition    by definition?

Amended:

Line 275:

The resulting unrelated individuals are by definition distantly related individuals

L232   I do not really understand what you mean with indirect genetic effects?

Amended:

Line 281:

Indirect genetic effects may result from genetic maternal effects [28].

L265    First time that you talk about the phenotype as a binary trait. Should be done in Materials and methods

Amended:

See above. Lines 125-149

L269   Also, you discuss here how you selected animals but this needs to be described in the M&M

Amended:

See above. Lines 125-149

L283 …which may also could have had….sounds very heavy….which could have had….

Amended:

Line 331:

proved version of the software, which could have had an effect on the results.

L241 to L259    You discuss contributions to the estimates but I do not see where we can find these results. h2SNPf  is something you introduce her but was not mentioned before. This makes this part of the discussion a bit confusing.

Amended:

Lines 171-179:

Using the GREML analysis for family data with two GRMs simultaneously, based on all pairs of individuals and related individuals [24], implemented in GCTA 1.94.1 [30], we estimated .  The GRM based on all pairs of individuals captures information on sharing of causal variants that are tagged by SNPs. The second GRM considers only individuals who are identical-by-state above a certain threshold (0.05) and, consequently only related individuals. Hence, it captures information on shared causal variants that cannot be tagged by SNPs [24,29]. Both GRMs are fitted into a mixed linear model and are supposed to provide estimates of  from the  first GRM and the missing heritability  from the second GRM. Those values are summed up to  [24]

L296    Conclusions

I’m not convinced that a pedigree based approach would yield lower accuracy. Maybe you can test it on your data and run a pedigree-based analysis.

Amended:

Lines 345-347

Estimation of heritabilities based on SNP-arrays is recommended because reasonably high accuracy of estimates can be achieved in smaller samples compared to pedigree-based studies with similar sample sizes.

I think however that the fact that imputation is maybe not needed and that family based data can be used is more important for the horse community.

Amended:

Lines 351-353

We deleted the sentence.

Supplementary figure 1.

The complete figure is not really needed. The number of retained SNP’s in original and imputed datasets, after pruning for MAF, is certainly informative but the horizontal, colored bars are not needed.

Amended: We kept this figure to avoid misunderstandings.

Table S1:  Estimates of heritability for equine osteochondrosis reported in previous studies

The table does not seem to be very complete. This are not all previous studies. But I can understand that you do not want to write a review paper (again) but than you need to give an explanation why these studies and not other studies.

Amended:

The Supplementary Table was updated. Additionally, we decided to cite the already published review instead of showing the same results in our table repeatedly.

Amended:

Supplementary Table 1

Lines 53-57

There have been estimations regarding the heritability of osteochondrosis and osteochondrosis dissecans based on pedigree data [5-21] as well as on genotyping data [22,23]. Those estimates are shown in a previous review [1] and are supplemented by results of more recent studies in Supplementary Table S1.

Reviewer 3 Report

The paper is very intersting, howeve rit should be clarified in many aspects:

1. the trait is not described, some earlier study is cited, however it is written that only a part of horses was included - what was the base of inclusion? what is the trait for which h2 is calculated? the ocd scale-trait, definition used for this study is not given. The h2 depends on trait definition. Please describe the trait, data inclusion method, horses used . 

2. it seems that the statistical model is not correct (environmental factors missing?). the h2 is above 1, it is not the information for the practise.

3. the number of horses is not high - perhaps short communication would fitt better for this paper 

4. the preselection effect is not dicussed

There are some others mistakes that could be corrected, but first the main trait definition and material describtion should be clarified. 

Author Response

Reviewer 3:

We thank the reviewer for the valuable comments and recommendations to improve our manuscript. We revised our manuscript accordingly.

Open Review

Quality of English Language

( ) English very difficult to understand/incomprehensible
( ) Extensive editing of English language and style required
( ) Moderate English changes required
( ) English language and style are fine/minor spell check required
(x) I am not qualified to assess the quality of English in this paper

Yes

Can be improved

Must be improved

Not applicable

Does the introduction provide sufficient background and include all relevant references?

(x)

( )

( )

( )

Are all the cited references relevant to the research?

(x)

( )

( )

( )

Is the research design appropriate?

( )

(x)

( )

( )

Are the methods adequately described?

( )

( )

(x)

( )

Are the results clearly presented?

( )

( )

(x)

( )

Are the conclusions supported by the results?

( )

(x)

( )

( )

Comments and Suggestions for Authors

The paper is very intersting, howeve rit should be clarified in many aspects:

  1. the trait is not described, some earlier study is cited, however it is written that only a part of horses was included - what was the base of inclusion? what is the trait for which h2 is calculated? the ocd scale-trait, definition used for this study is not given. The h2 depends on trait definition. Please describe the trait, data inclusion method, horses used . 

Amended:

Lines 125-149

The inclusion criteria were as follows: only one horse per sire and maternal sire was allowed, either as a control or a case. The controls and cases were randomly distributed among the sires and maternal sires. The control horses had to be free of all diseases found during the veterinary health examination for pre-selection at auctions, at licensing, or during the purchase examination. The cases were horses with OCD only and free of any other disease recorded in the veterinary health check. The veterinary health check included clinical and radiographic examination of all four limbs. Only osteochondral fragments at the specific predilection sites of the fetlock, hock, and stifle joints were classified as OCD [5]. Osteochondral fragments plantar in the fetlock joints and at the insertion sites of the short sesamoid ligaments at the proximal phalanx of the hindlimbs were classified as plantar and dorsodistal fragments of the fetlock joints and were not considered OCD. Distal and proximal interphalangeal joints, fetlock joints, hock joints, and stifle joints were evaluated for contour changes caused by periarticular osteophytes or exostoses and for a narrow or absent joint space. These changes were classified as osteoarthroses. Radiographic changes in the shape, symmetry, contour, and structure of the navicular bone and the shape, size, number, and location of the canales sesamoidales were scored on a scale of 1-4. Only horses with a score of 1 were considered free of radiographic changes of the navicular bone. The presence of a sidebone was also scored as not free of radiographic changes. After removing all horses affected by diseases other than OCD, the data set was filtered for cases and controls. The strict inclusion criteria resulted in the final data set comprising 446 horses.

  1. it seems that the statistical model is not correct (environmental factors missing?). the h2 is above 1, it is not the information for the practise.

Lines 314-315:

In agreement with previous studies, upward bias may occur, particularly, when estimates on the linear scale are high and the more frequencies deviate from 0.5 (11,12).

Amended:

Line 181:

Sex was included in all analyses as a covariate. Other effects were not significant.

  1. the number of horses is not high - perhaps short communication would fitt better for this paper 

We think that we present a significant advance in analysis of SNP-based heritabilities.

  1. the preselection effect is not dicussed

Amended:

Outcome of previous papers was: We assume that sampling of horses for pre-selection for sale at auction or pre-selection for stallion licensing or other reasons did not influence the estimates of heritabilities.

We did not discuss this matter because this topic has already been discussed in all previous papers.

There are some others mistakes that could be corrected, but first the main trait definition and material describtion should be clarified. 

Amended:

Mistakes amended.

Round 2

Reviewer 1 Report

Authors completed the manuscript with the suggested details

Reviewer 2 Report

Thank you for considering my questions and suggestions. 

You made many things more clear.

Reviewer 3 Report

Thank you for your review.